# A Rare Case of Anca Positivity and Antiphospholipid Antibodies in a Patient with Takayasu Arteritis: Case Report and Review of the Literature

**DOI:** 10.3390/biomedicines11102826

**Published:** 2023-10-18

**Authors:** Rositsa Dacheva, Ekaterina Kurteva, Vladimira Boyadzhieva, Rumen Stoilov, Dobroslav Kyurkchiev, Nikolay Stoilov

**Affiliations:** 1Department of Rheumatology, Clinic of Rheumatology, University Hospital St. Ivan Rilski, Medical University of Sofia, 1612 Sofia, Bulgaria; vladimira.boyadzhieva@gmail.com (V.B.); rmstoilov@abv.bg (R.S.); dr_nstoilov@abv.bg (N.S.); 2Laboratory of Clinical Immunology, Department of Clinical Immunology, University Hospital St. Ivan Rilski, Medical University of Sofia, 1431 Sofia, Bulgaria; dsk666@gmail.com

**Keywords:** antiphospholipid antibodies, Takayasu arteritis, ANCA

## Abstract

Takayasu arteritis (TA) is a chronic large-vessel vasculitis characterized by immune-mediated panarteritis, which predominantly affects the aorta and its main branches and is most prevalent in young women. TA is unusually associated with the presence of antiphospholipid antibodies. We present a case report of a 48-year-old Caucasian woman with acute aortic dissection as an initial feature of TA, where detailed clinical, imaging and laboratory studies were performed. Computed tomography angiography (CTA) of the chest and abdomen revealed aortic dissection DeBakey I. Bentall and De Bono surgery was performed. Additional immunological tests revealed positive antineutrophil cytoplasmic antibodies (ANCAs) with the simultaneous presence of pANCA and cANCA antibodies on indirect immunofluorescence, along with anti-MPO+PR3+antibodies positivity in the absence of a clinically relevant disease. Surprisingly, antiphospholipid antibodies (aPLs) were detected. Then, we performed a thorough review of the current literature. The coexistence of aPL antibodies and dual specificity for MPO and PR3 in a patient diagnosed with Takayasu arteritis is unusual and poses a diagnostic challenge. The presented case report outlines a rare case of aortic dissection as a presenting symptom of TA, along with atypical ANCA positivity and positive APL antibodies.

## 1. Introduction

Takayasu arteritis (TA) is a chronic idiopathic large-vessel vasculitis characterized by immune-mediated inflammation primarily affecting the aorta and its main branches. It is also called “pulseless disease” and affects predominantly young women. The pathogenesis consists of the involvement of all arterial layers (panarteritis). The inflammatory process may result in stenosis, occlusion or aneurysm formation in the involved arteries [1,2].

The diagnosis of TA is complex and is based on systemic and vascular symptoms in combination with imaging results. The 1990 American College of Rheumatology (ACR) Classification Criteria for TA [3] is widely used in routine clinical practice. The disease can be accepted if three or more of the following criteria are met: (1) age at disease onset <40 years; (2) claudication of extremities; (3) decreased brachial artery pulse; (4) blood pressure difference >10 mm Hg between arms; (5) bruit over subclavian arteries or aorta; and (6) arteriogram abnormality [4]. Diagnostic imaging is fundamental in diagnosing Takayasu arteritis and is essential in monitoring the disease [5]. In 2022, the new classification criteria for TA were published. The novel criteria incorporate the advancement in imaging techniques, including evidence of vasculitis on imaging as an absolute requirement. Another obligatory criterion is the age at onset ≤60 years. Additional criteria are based on a scoring system including clinical features, such as female sex, angina or ischemic cardiac pain, arm or leg claudication, vascular bruit, reduced pulse in the upper extremity, carotid artery abnormality and systolic blood pressure difference in arms ≥20 mm Hg, and imaging criteria, including the number of affected arterial territories, symmetric involvement of paired arteries and abdominal aorta involvement. Patients classified as having a cumulative score ≥ 5 will receive a TA diagnosis [6]. According to the site of involvement, TA was classified by Hata A et al. into five types: type I, branches of the aortic arch; type IIa, ascending aorta, aortic arch, and its branches; type IIb, ascending aorta, aortic arch and its branches, and thoracic descending aorta; type III, thoracic descending aorta, abdominal aorta, and/or renal arteries; type IV, abdominal aorta and/or renal arteries; and type V, combined features of types IIb and IV. According to this classification system, the involvement of the coronary or pulmonary arteries should be designated as C (+) or P (+), respectively [5]. Vessel stenosis is present in 85% or more of patients at diagnosis and is the most common arteriographic finding. Aneurysms may be saccular or fusiform and typically affect the aorta rather than its branches [5]. Aortic dissection is a rare complication of TA, and only a limited number of TA cases presenting with aortic dissection have been published so far [7,8,9]. The initial site of the inflammatory process is the adventitia, progressing to the intima and finally evolving to panarteritis. The tissue damage resulting from the inflammation heals with fibrosis [10]. The lack of specific laboratory pathognomonic indicators for the disease also hampers the diagnostic process. There are no specific antibodies associated with TA. Moreover, antineutrophil cytoplasmic antibodies (ANCA) have been rarely reported in patients with TA.

The current case report represents a female patient diagnosed with TA expressing antibodies simultaneously against myeloperoxidase (MPO) and proteinase-3 (PR3). In addition, antiphospholipid antibodies (aPLs) were also detected.

## 2. Case Presentation

On the 14 August, a 49-year-old Bulgarian female was admitted to the Cardiac Surgery Department of the University Hospital City Clinic, Bulgaria, with complaints of severe chest pain with sudden onset starting on the 12 August. The patient reported persistent pain in the jaw and subsequent palpitations for two days. Clinical examination revealed that the woman was in a moderately impaired general condition with a blood pressure of 115/60 and a heart rate of 70 per minute. A diastolic heart murmur was detected at the right upper sternal border. Laboratory tests showed increased C-reactive protein (CRP) with other laboratory parameters within the reference range. Echocardiography revealed aortic arch dissection, aortic valve regurgitation, preserved size and left and right ventricle volumes with preserved ventricular function. Computed tomography angiography (CTA) of the chest and abdomen revealed aortic dissection DeBakey I (Figure 1). A variety of origins of supra-aortic vessels—bovine arch was detected. There was a flap in the initial truncus brachiocephalicus segment and the ostium of the left common carotid artery. There was thrombosis in the false lumen in the initial segment of the left common carotid artery. There was a separation of the left subclavian artery from the false lumen, with a flap in the ostium. Bentall and De Bono surgery was performed with an uneventful postoperative period. Anticoagulant and antiplatelet therapy was initiated, including acenocoumarol and clopidogrel.

The patient was referred to a rheumatologist followed by an admission to the Clinic of Rheumatology for further evaluation and treatment. On physical examination, the patient presented with a systolic blood pressure difference >10 mmHg between arms. The patient had no history of hypertension. The musculoskeletal examination showed no pathological findings. The laboratory investigation revealed elevated CRP. Blood count, liver enzymes, electrolytes and urine tests were in the normal reference range. Initial laboratory and follow-up results are depicted in Table 1. The patient reported a history of transient thrombocytopenia and one spontaneous abortion before 10 weeks of gestation. For those reasons, we ordered thorough immunologic tests, including aPL.

Immunologic examination disclosed positive ANCA in high titres 1:640 with negative value < 1:20, positive anti-proteinase three antibodies (anti-PR3) > 100 and anti-myeloperoxidase antibodies (anti-MPO) > 100 with negative value < 5 and positive antiphospholipid anti-bodies (anticardiolipin ACL-screen—29.60 U/mL with negative value < 10; anti-b2GPI—22.30 U/mL with negative value < 10; anti-prothrombin 67.10 U/mL with negative value < 20). Antinuclear antibodies (ANA) tested by immunofluorescence assay (IFA) and ANA-immunoblot test were negative. Immunologic tests are shown in Table 2. An IGRA test and a screening test for hereditary thrombophilia were performed with negative results. The patient had no family history of connective tissue disorders. The tests performed did not reveal ear, nose, throat (ENT) or kidney involvement, or any relevant organ damage characteristic for ANCA-associated vasculitis. A diagnosis of TA was established, and the patient was referred for cardiac surgery for further testing and biopsy. Immunosuppressive therapy was administered in an initial daily dose of 100 mg of Azathioprine and 400 mg of Hydroxychloroquine. Follow-up examination after six months revealed similar immunological results, with confirmed positive values for aPL and positive high-titer ANCA (1:640) throughout follow-up with double positivity for anti-MPO and anti-PR3. Further assessments would evaluate the effectiveness of the therapy and would determine the need for treatment escalation.

## 3. Materials and Methods

### Search Strategy

A literature review was conducted according to the accepted rules [11]. Medline and Scopus databases were searched using the following keywords “Takayasu arteritis” or “antiphospholipid antibodies” or “antineutrophil cytoplasmic antibodies” or “anti-PR3” or “anti-MPO”. Two reviewers (EK and RD) independently reviewed all of the articles in full text. In addition, we screened their references for appropriate inclusions. We selected case-reports, systematic reviews, recommendations and primary studies published in a peer-reviewed journal, providing empirical data concerning Takayasu arteritis and presence of aPL, ANCA or atypical course of the disease. There were no disagreements between the two reviewers during the selection process. The flowchart of the literature selection process is depicted in Figure 2.

ANCA pattern is depicted in Figure 3.

## 4. Discussion

### 4.1. Simultaneous Positivity against MPO and PR3

ANCA are formed in response to a number of cytoplasmic or perinuclear antigens in neutrophil, with the most prevalent being myeloperoxidase (MPO) and proteinase 3 (PR3) [12]. MPO and PR3 are detectable in the blood serum of healthy individuals due to neutrophil activation or death [13].

ANCA-associated vasculitis comprises a heterogeneous group of autoimmune diseases affecting small blood vessels. Subclassifying AAV relies on the subtype of ANCA and the determined treatment decisions and disease outcomes. ANCAs have a great diversity of targets, PR3 and MPO being the two major and most recognized antigens with distinct clinical significance. Although characteristic of AAV, ANCAs have been reported in the blood serum of healthy individuals. Alternative diagnoses in which ANCA positivity was previously reported include non-vasculitides inflammatory diseases, systemic lupus erythematosus, seronegative rheumatoid arthritis, giant cell arteritis, sarcoidosis, systemic sclerosis, Crohn’s disease and systemic sclerosis [14]. The patients expressing antibodies simultaneously against MPO and PR3 (anti-MPO+PR3+) are not frequently detected. A retrospective study concerning ANCA-associated vasculitis demonstrated that out of 856 patients, only 8 showed antibodies against MPO and PR3. Six of them were diagnosed with MPA, and two with GPA. Moreover, anti-MPO+PR3+ patients are characterized by renal involvement, which approaches them to anti-MPO-positive vasculitis and demonstrates upper respiratory tract involvement, characteristic of anti-PR3-associated vasculitis [15]. Other studies also support the low incidence of concomitant anti-PR3 and anti-MPO in ANCA-associated vasculitides [16,17]. This immunological constellation may also be seen in the overlap between ANCA-associated vasculitis and mixed connective tissue disease. In addition, the simultaneous staining pattern of pANCA and cANCA on indirect immunofluorescence, along with the presence of anti-MPO+PR3+antibodies, is found in diseases, except for the group with AAV, like lupus nephritis [18], drug-induced nephritis [19], Schönlein–Henoch disease [20], and infectious endocarditis [21,22]. However, it must be noted that these reports represent predominantly isolated clinical cases [23]. In a study attempting to describe the association between the simultaneous positivity for anti-PR3 and anti-MPO antibodies and existing disorders, 15 patients with various diseases were identified, some of which were not even autoimmune by nature, such as solid tumors and infections [24]. Choi et al. determined the frequency of ANCA positivity in patients with TA. Out of 121 patients with TA, 8 (6.6%) had a positive ANCA, of which 2 patients had positive result for both MPO and PR3 [25]. Dual MPO- and PR3-ANCA specificities in SLE patients have also been observed by Pradhan et al. [26]. A recent report identified a patient with dual positive MPO- and PR3-AAV following SARS-CoV-2 mRNA vaccination [27].

### 4.2. Antiphospholipid Antibodies and Takayasu Arteritis

aPLs are associated with venous and arterial thrombosis, thrombocytopenia and obstetric complications. The implication of aPLs in TA is controversial since the vasculopathy associated with secondary antiphospholipid syndrome (APS) affect predominantly small or medium-sized vessels with vasculopathy being primary thrombotic [28]. The main mechanism associated with APS is thrombosis leading to a secondary vasculopathy. However, vasculitis with inflammation of the vessel wall being the primary site of involvement is also described as a distinct manifestation of APS. Differentiating between vasculitis and vasculopathy may pose a challenge to the clinician.

TA generally affects young women, with a female-to-male ratio of 8.5 to 1, and disease onset in most cases is before 40 years [29]. The association between APS and TA has been reported in the literature. Positive aPL prevalence in Takayasu patients varies between 0% and 53% [18,19,20,21]. A retrospective study by Jordan et al. demonstrated that 41% (9/22) of women enrolled had persistently positive aPLs or were diagnosed with concurrent antiphospholipid syndrome. The aPL positivity in TA concurs with antiendothelial cell antibodies and disease activity, such as in the case of Giant-cell arteritis [30,31]. In Takayasu, aPLs could be considered an epiphenomenon reflecting the extent of vascular endothelial impairment, although some research teams speculate that aPLs may contribute to the late obstructive vasculopathy in Takayasu following the initial inflammatory phase. Positive anticardiolipin antibodies (ACLAs) were observed in several reports [27,32,33]. Haviv reports a possible association and a pathogenetic role of ACLAs in vascular injury triggered by endothelial activation and immunological or apoptotic processes [34]. Tripathy et al. investigated 66 TA patients and 50 healthy controls for anti-annexin V antibodies (AAV5As), ACLAs and antiendothelial cell antibodies, revealing 24 positive results for AA5As compared to 3 out of 50 healthy controls [35]. Şentürk et al. have investigated 49 females who received a TA diagnosis. Nine out of forty nine (16.9%) patients had positive β2GP1 or lupus anticoagulant (LA) antibodies in low titer. No patients reported a history of venous thrombosis in the aPL (+) group and only one female with IgM β2GP1 (+) had a history of pregnancy loss [36].

## 5. Conclusions

TA is a large-vessel vasculitis manifesting with inflammation predominantly affecting the aorta and its branches. Diagnosis of TA remains challenging for the clinicians due to the diverse clinical manifestations of the disease ranging from subclinical features to life-threatening events. In general, the appearance of aortic dissection as an initial manifestation of TA is uncommon. Furthermore, although the presence of antibodies is not characteristic of TA, the presented clinical case describes a rare coexistence of multiple antibodies, distinctive for separate diseases with diverse pathogenesis and clinical features. The presence of positive ANCA antibodies with dual specificity for MPO and PR3 in high titers and concomitant presence of aPL antibodies is undoubtedly a finding of interest. The results have been confirmed in multiple serial examinations ruling out the possibility that this is an incidental finding. In addition, the patient had no apparent clinical manifestation associated with ANCA antibodies. The occasional presence of ANCAs or aPLs in patients diagnosed with TA has been previously reported in the literature. However, whether the abovementioned antibodies have pathogenetic role or are epiphenomenon of the disease needs further evaluation.

## Figures and Tables

**Figure 1 biomedicines-11-02826-f001:**
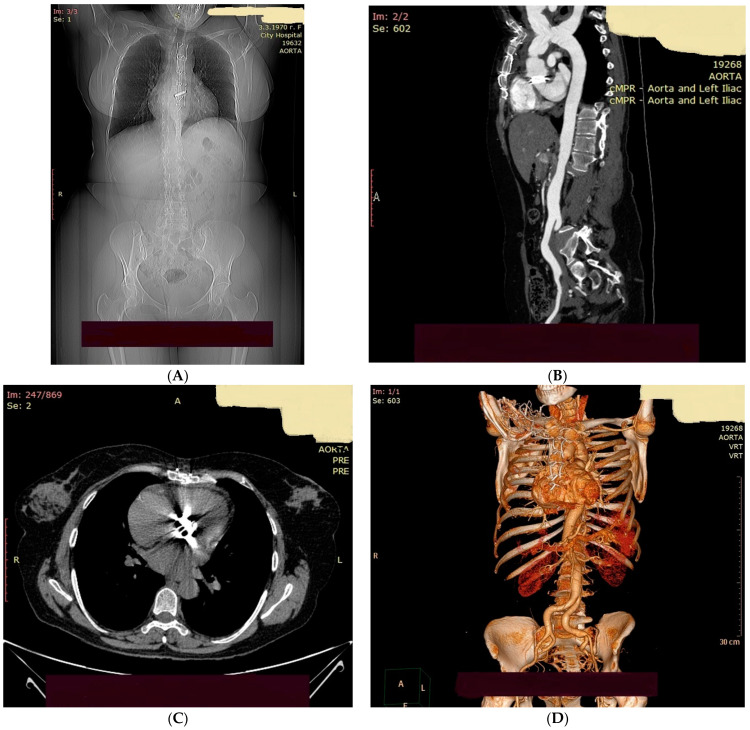
(**A**) Status post aortic mechanical valve replacement. (**B**) Ascending aortic prosthesis and coronary ostium reimplantation were performed to correct Debakey I aortic dissection. (**C**) A dissection flap is traceable in the aortic arch and distally in the initial segment of the descending thoracic aorta. (**D**) After stent implantation in the left common carotid artery, supra-aortic vessels are contrasted with descriptions of varieties and available dissection flaps. Dissection of the common iliac arteries is visualized; mild mediastinal lymphadenomegaly is present.

**Figure 2 biomedicines-11-02826-f002:**
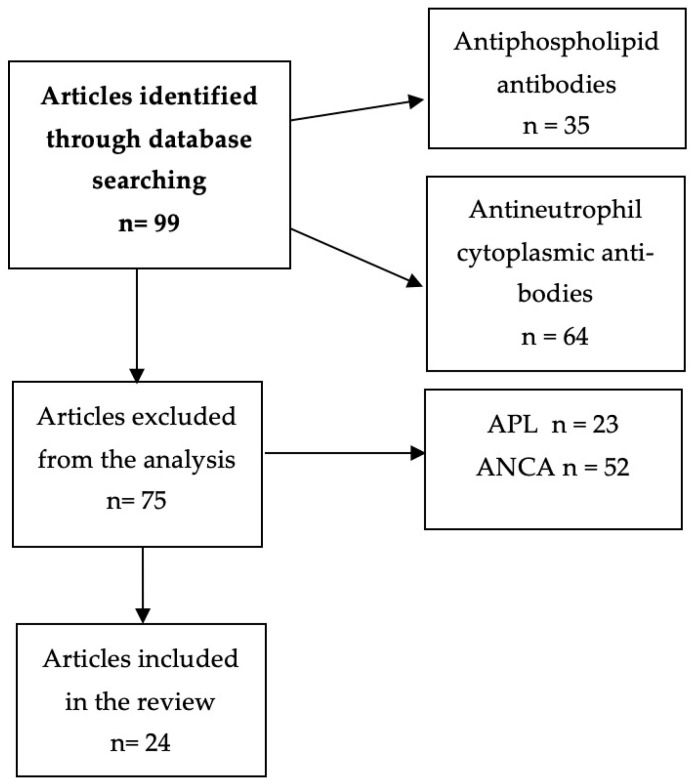
Article selection process flowchart. APL—antiphospholipid antibodies. ANCA—antineutrophil cytoplasmic antibodies.

**Figure 3 biomedicines-11-02826-f003:**
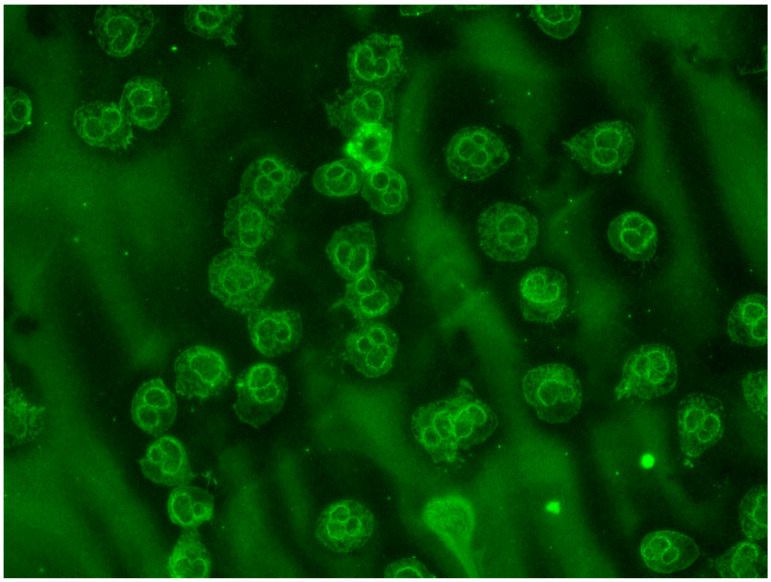
ANCA pattern of the patient with double positivity for anti-MPO and anti-PR3.

**Table 1 biomedicines-11-02826-t001:** Initial laboratory results.

Laboratory Variable	Result	Follow-Up Results	Reference Rage
White blood cells	5.72	5.72	3.5–10.5 × 10^9^/L
Red blood cells	5.22	5.22	3.7–5.3 × 10^12^/L
Hamoglobin	132	137	115–150 g/L
Hematocrit	0.397	0.399	0.350–0.490
Platelet	156	96	112–330 × 10^9^/L
Erythrocyte sedimentation rate	17	21	<20 mm/h
NEUT %	64	64.3	38–78%
LYM %	26.7	26.2	15–41.9%
C-reactive protein	15.7	2.0	<6 mg/L
Creatinine	72	75	Up to 96 µmol/L
Urea	5.50	5.4	1.7–8.2 mmol/L
Uric acid	257	262	142–340 µmol/L
ASAT	18.5	29.10	Up to 32 U/L
ALAT	14.10	28.20	Up to 33 U/L
Creatine kinase	80		24–170 U/L
GGT	22.0	24	Up to 40 U/L
Fibrinogen	4.36	4.25	2.0–4.0 g/L

**Table 2 biomedicines-11-02826-t002:** Immunology laboratory results.

Immunologic Test	Result	Follow-Up Results	Reference Range
ANA-immunofluorescence assay	1:160	negative	<1:160
ANA immunoblot	Normal		
Antineutrophil cytoplasmic antibodies	1:640	1:640	<1:20
Anti-PR3	>100	13.80	<5
Anti-MPO	>100	31.80	<5
Anticardiolipin antibodies	29.6	13.80	<10
Beta-2 Glycoprotein 1 antibodies	22.3	31.80	<10
Anti-protrombin antibodies	67.10	23.60	<20
Complement C3	1.40		0.81–1.57 g/L
Complement C4	0.38		0.13–0.39 g/L

## Data Availability

No new data were created.

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
