# Peer review of "A Rare Case of Anca Positivity and Antiphospholipid Antibodies in a Patient with Takayasu Arteritis: Case Report and Review of the Literature"

_biomedicines, 2023, doi:10.3390/biomedicines11102826_

Round 1

Reviewer 1 Report

The article "A RARE CASE OF ANCA POSITIVITY AND ANTIPHOSPHOLIPID ANTIBODIES IN A PATIENT WITH TAKAYASU ARTERITIS: CASE REPORT AND REVIEW OF THE LITERATURE" describes a case of Takayasu arteritis positive for both aPL (anti-cardiolipin antibodies) and ANCA. The authors also report a review of the literature.

The article is not in-depth and summarily reports both the clinical case presented and the few reviews already done on the associations between Takayasu arthritis with both aPL and ANCA.

The few data presented do not bring new scientific findings, except for a new case of takayasu arteritis in association with aPL and ANCA. The clinical case is not asolutely in-depth and remains mere data. The literature review does not allow for added insights into the scientific and research field of large vessel vasculitis and ANCA-associated vasculitis.

The work presented in this way is not acceptable for the purposes of the journal. It can be referred as a case report to a rheumatology journal.

Reviewer 2 Report

I read this manuscript with interest. These are my comments.

1. I agree that this is an interesting case report. However, there are many minor errors throughout the manuscript. For example,

Line 64, pregressing→progressing

Line 161, PR+ ➡PR3+

Line 190, Takayaus → Takayasu arteritis or TA, etc.

Please read and revise throughout the entire manuscript.

2. lab data

Please make a table of the lab data. 

3. aPL in Takayasu arteritis

As the authors cite, up to 53% of patients with Takayasu arteritis have aPL, which is repeatedly described as rare, but may not be rare. The authors need to tone down a bit.

4. clinical course

Please show the clinical course in Figure. How did the titers of ANCA change?

 there are many minor errors throughout the manuscript.

Round 2

Reviewer 1 Report

Nothing have been changed with respect to the previous version.

We are still on our opinions.

Author Response

We greatly appreciate your comments.

Reviewer 2 Report

All the questions raised by this reviewer were addressed accordingly.

Author Response

Thank you for the opportunity to submit a revised draft of the manuscript. We appreciate your valuable comments.